# DELTA - CONTRASTIVE DECODING MITIGATES TEXT HALLUCINATIONS IN LARGE LANGUAGE MODELS

## ABSTRACT

Large Language Models (LLMs) have demonstrated remarkable capabilities in natural language processing tasks. Still, they are prone to generating hallucinations—factually incorrect or fabricated content that can undermine their reliability, especially in high-stakes domains such as healthcare and legal advisory. In response to this challenge, we propose Delta, a novel inference-time approach that leverages contrastive decoding to mitigate hallucinations without requiring model retraining or additional training data. Delta works by randomly masking portions of the input prompt, then contrasting the original and masked output distribution generated by the model, effectively mitigating hallucinations through inference-only computations. Delta was evaluated on context-rich QA benchmarks like SQuAD v1.1 and v2, achieving around 3 and 6 percentage points of improvement, respectively. It also showed gains of 7 and 2 percentage points on TriviaQA and Natural Question under-sampling decoding. Delta improved SQuAD v2's no-answer exact match by over ten percentage points. These findings suggest that Delta is particularly effective when hallucinations arise from contextual ambiguity. Delta presents a computationally efficient and scalable solution for reducing hallucinations in real-world LLM applications by focusing on inference-time enhancements.

## 1 INTRODUCTION

The rapid development of large language models (LLMs) Brown et al. (2020) has led to significant advancements in text generation, natural language processing, and a wide range of real-world applications OpenAI et al. (2024). These models, powered by vast datasets and complex architectures, have become essential tools for translation, summarization, and conversational AI tasks McKenna et al. (2023). Despite these achievements, LLMs face a critical challenge: their probabilistic and nondeterministic nature often generates "hallucinated" content Xu et al. (2024). These hallucinations manifest as text that may sound plausible but is factually incorrect or fabricated. This poses a significant issue, particularly in high-stakes domains such as healthcare, legal advisory, and scientific research, where the accuracy and reliability of the generated content are paramount.

Hallucinations in LLMs arise from their reliance on patterns learned during training, causing them to occasionally generate outputs unsupported by the input data or real-world facts Huang et al. (2023). Addressing this issue is crucial for improving the reliability and trustworthiness of LLMs, especially as they are increasingly integrated into real-time systems and applications where incorrect information can lead to severe consequences. In response to this challenge, we introduce Delta, a novel approach designed to identify and mitigate text hallucinations on inference-time computation. Unlike traditional methods Ji et al. (2023); Li et al. (2023b); Ouyang et al. (2022) focusing on retraining models or requiring access to additional data, Delta operates solely at inference time, making it computationally efficient and easily deployable in real-time systems. Delta's core innovation lies in its use of contrastive decoding Li et al. (2023a); Chuang et al. (2024), which leverages masked versions of the input text to contrast plausible outputs against potentially hallucinated ones.

Delta builds upon previous work in vision-language models, particularly the method introduced in Leng et al. (2024), which mitigates object hallucinations by applying Gaussian noise to the visual input during contrastive decoding. However, adapting this approach to text is more challenging, as directly applying noise to textual input is not feasible. To address this, Delta employs a masking

strategy Wettig et al. (2023), where tokens in the input sequence are randomly masked to simulate ambiguity. Delta can filter out hallucinated content more effectively during inference by comparing the model's predictions on masked versus unmasked inputs.

Our experimental results demonstrate the effectiveness of the Delta method, achieving notable improvements in question-answering accuracy. Specifically, Delta delivered gains of approximately 3 and 6 percentage points on SQuAD v1.1 and v2 Rajpurkar et al. (2016), respectively. It achieved a 14.53 percentage point improvement in the no-answer exact match score on SQuAD v2. For more challenging QA datasets like TriviaQA Joshi et al. (2017) and Natural Questions Kwiatkowski et al. (2019), Delta achieved enhancements of 7 and 2 percentage points under-sampling decoding compared to the baseline. These results confirm Delta's robustness and effectiveness in context-rich datasets, showcasing its ability to mitigate hallucinations and enhance performance.

However, a fundamental limitation of Delta is its marginal effectiveness on tasks without explicit contextual information. For instance, datasets like CommonsenseQA Talmor et al. (2019) and MMLU, which rely heavily on general or implicit knowledge, showed minimal or no improvements, indicating the method's specific suitability for context-driven scenarios.

## 2 RELATED WORKS

Recent studies have focused on mitigating hallucinations in large language models (LLMs) and large vision-language models (LVLMs) Hinck et al. (2024), where models tend to generate inaccurate or irrelevant outputs. In vision-language models, such hallucinations often result from over-reliance on language priors or biases embedded in the datasets. For example, in LVLMs, object hallucinations occur when models predict objects not present in the image due to biased object co-occurrences in the training data. Several approaches have been developed to address this, including Visual Contrastive Decoding (VCD) and Instruction Contrastive Decoding (ICD).

The *Visual Contrastive Decoding (VCD)* method aims to reduce object hallucinations by contrasting the outputs generated from original and distorted visual inputs. This approach does not require additional training or external pre-trained models, making it a computationally efficient solution. By introducing visual uncertainty, such as Gaussian noise, the method identifies and mitigates instances where the model overly relies on language priors or statistical biases from the training data, thereby reducing hallucinations Leng et al. (2024).

Similarly, the *Instruction Contrastive Decoding (ICD)* technique is employed to tackle hallucinations in multimodal tasks by incorporating instruction disturbances. This method manipulates the confidence of multimodal alignment in the model's visual and textual inputs, helping it differentiate between hallucinated and relevant tokens. By applying contrastive penalties to tokens influenced by instruction disturbances, ICD effectively reduces the generation of hallucinated outputs, especially in complex visual contexts Leng et al. (2024).

In addition, the work context-aware decoding (CAD) has Shi et al. (2024) demonstrated a similar outcome to our Delta method by adjusting the output probabilities of LMs, amplifying the differences between outputs generated with and without the given context. This contrastive approach encourages models to prioritize contextual information during text generation. Notably, CAD can be applied to pre-trained LMs without additional training. Unlike our Delta method, the method is mainly based on context-driven datasets, making it less generalizable than the Delta method, which, in theory, could apply to all textual inputs.

Both approaches are part of a growing body of work exploring contrastive mechanisms and fine-grained multimodal alignment techniques to mitigate hallucinations in models that integrate vision and language processing. Future research will likely explore more robust mechanisms further to improve model reliability across different types of multimodal tasks.

## 3 METHOD

The work introduces Delta, a novel method that effectively mitigates hallucinations in text-based large-language models. The core idea behind Delta is to address the issue of hallucinations by manipulating the inference process itself. Specifically, the method generates output tokens from the

model using a standard inference procedure, as outlined in Equation 1. Building on hypotheses inspired by Leng et al. (2024), which suggest that incomplete prompting or missing information tends to amplify hallucination effects, Delta aims to mitigate this by leveraging a contrastive decoding approach.

Delta dynamically adjusts for incomplete information in this approach by comparing the outputs generated from masked and unmasked input versions. The key concept behind Delta is as follows: by randomly masking input tokens, the model generates outputs that are more likely to be filled with hallucinated information. Then, by subtracting the hallucinated logits (generated from the masked input) from the original logits, Delta extracts the "clean" logits—those less influenced by hallucinated content. This process significantly reduces the likelihood of hallucinations, as demonstrated in Figure 1, leading to more accurate and reliable outputs in context-dependent tasks.

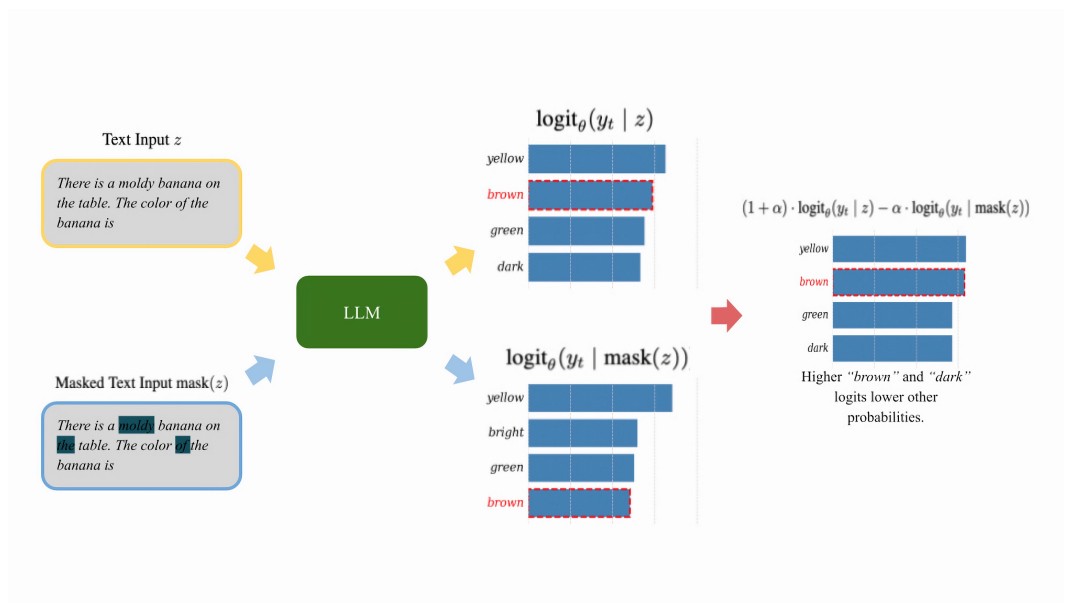

Figure 1: Illustrating Delta method by contrastive decoding with masked input prompting

## 3.1 INFERENCE FROM LANGUAGE MODEL DECODER

In large language models (LLMs), the inference process involves predicting the next token in a sequence based on the previously generated tokens. Given an input sequence $x$ and generated tokens $y$, we can have $z = [x_0, x_1, \ldots, x_{n-1}, y_1, \ldots, y_{t-1}]$ where $n$ is the index of the sequence, the conditional probability of the next token $y_t$ at time step $t$ is modeled as:

$$P_\theta(y_t \mid z) = \text{softmax}\left(\text{logit}_\theta(y_t \mid z)\right) \tag{1}$$

In this equation, the model generates tokens sequentially, where the logits are computed using the model's parameters $\theta$. This process is crucial for autoregressive tasks like text generation, where each token is conditioned on all the tokens that came before it.

## 3.2 EFFECT OF FUZZIFIED (MASKED) TEXT ON HALLUCINATIONS

Masking portions of input text in large language models can exacerbate hallucinations. For instance, consider the sentence: "There is a moldy banana on the table. The color of the banana is "". If the word "moldy" is masked and replaced with the token "MASK," the sentence becomes "There is a MASK banana on the table. The color of the banana is." In this case, a model that relies heavily on its pre-trained knowledge may output a high logit value for "yellow," a common association for bananas, even though "brown" would be the more accurate color based on the original context. This

behavior highlights how the model might default to its prior associations when deprived of important contextual cues, leading to an output that is factually incorrect or "hallucinated."

The issue stems from the model's tendency to draw upon its pre-trained knowledge to fill the missing context. Without access to specific context or visual input, the model resorts to typical patterns learned during pre-training, such as associating bananas with the color "yellow," a more frequently seen context in training data. This reliance on default associations can result in outputs that are not grounded in the given input but instead reflect generalized patterns, which may not always be accurate. As illustrated in Figure 1, this effect occurs when the masking introduces ambiguity or removes essential information, leading the model to generate plausible yet erroneous answers that fail to account for the specific context of the prompt.

### 3.3  TEXT SEQUENCE MASKING

In this process, we introduce ambiguity by randomly masking a portion of tokens within the input sequence. Given an input sequence $x = [x_0, x_1, \ldots, x_{n-1}]$, where $n$ is the length of the sequence, several tokens are replaced according to a predefined masking ratio. Specifically, the tokens to be masked are selected randomly, and the total number of masked tokens is determined by $m = \lfloor r_{\text{mask}} \cdot n \rfloor$, where $r_{\text{mask}} \in [0, 1]$ represents the masking ratio. The indices of the masked tokens are randomly selected and gathered into the set $I_{\text{mask}} = \{i_0, i_1, \ldots, i_m\}$.

The masked sequence is formalized in the following equation:

$$\text{mask}(x) = \left[x'_0, x'_1, \ldots, x'_{n-1}\right], x'_i = \begin{cases} \text{MASK} & \text{if } i \in I_{\text{mask}} \\ x_i & \text{otherwise} \end{cases} \tag{2}$$

The MASK token replaces the original tokens at the selected positions in this new sequence. The remaining tokens in the sequence remain unchanged. When such masked sequences are processed by large language models (LLMs), the model tends to predict tokens based on incomplete or missing context. This often leads to generating hallucinated words—words that are statistically likely based on the model's training data but not necessarily aligned with the original context.

### 3.4  CONTRASTIVE DECODING

The Delta method leverages contrastive decoding to enhance inference accuracy and reduce hallucinations in generated outputs. The key idea is to compare the predictions from masked and unmasked input sequence versions during token generation. At each time step $t$, the model generates the next token $y_t$ by conditioning on the unmasked sequence $z = [x_0, \ldots, x_{n-1}, y_1, \ldots, y_{t-1}]$ as well as its masked counterpart mask(z) where the $n$ is the index of the sequence $x$. The contrastive decoding process can be formalized in equation 3.

$$P_{\text{delta}}(y_t \mid z) = \text{softmax}\left[(1 + \alpha) \cdot \text{logit}_\theta(y_t \mid z) - \alpha \cdot \text{logit}_\theta(y_t \mid \text{mask}(z))\right] \tag{3}$$

In this equation, $\alpha \in [0, 1]$ (logit ratio) is a tunable hyperparameter that adjusts the relative significance of the masked logits. By subtracting the masked logits, which tend to induce stronger hallucinations, the method effectively reduces the influence of hallucinated token values in the original logits. The non-hallucinated tokens increase through $(1 + \alpha) \cdot \text{logit}_\theta(y_t \mid z)$. As the model prioritizes more plausible predictions from the unmasked context, this incrementation of probabilities for non-hallucinated tokens leads to a more accurate and reliable output. This implies that a higher $\alpha$ value can filter higher levels of hallucinations and amplify the level of non-hallucinated tokens.

### 3.5  ADAPTIVE PLAUSIBILITY CONSTRAINTS

To prevent the language model from generating imbalanced or semantically incorrect sequences. The work applied Adaptive Plausibility Constraints (APC) based on the Li et al. (2023a). The goal is to construct a set $V_{\text{head}}$ such that the logit with probability higher than the particular threshold, which is determined by the $\beta$, is not selected to the set $V_{\text{head}}$.

$$\mathcal{V}_{\text{head}}(x_{<t}) = \left\{ x_t \in \mathcal{V} : P_\theta(x_t \mid x_{<t}) \geq \beta \cdot \max_w P_\theta(w \mid x_{<t}) \right\} \tag{4}$$

Applying to APC, the language model could generate meaningful and semantically correct sentences even with the Delta method.

### 3.6 Computing Delta for Contrastive Decoding

Finally, the Delta method is computed during contrastive decoding. The core idea involves adjusting the logits for token generation by contrasting predictions from unmasked and masked versions of the input sequence. The following conditional equation defines how tokens are sampled at each time step $t$, represented by $y_t$:

$$y_t \sim \text{softmax}\left[(1 + \alpha) \cdot \text{logit}_\theta(y \mid z) - \alpha \cdot \text{logit}_\theta(y \mid \text{mask}(z))\right], \text{subject to } y_t \in \mathcal{V}_{\text{head}}(z_{<t}) \tag{5}$$

Here, the sequence $z$, which includes previously generated tokens, is checked against a set of plausible sequences $\mathcal{V}_{\text{head}}(x_{<i})$. If $z$ is deemed plausible, the model generates a token using modified logits, where the contribution of the unmasked sequence is amplified by a factor of $(1 + \alpha)$, and the contribution of the masked sequence is penalized by a factor of $\alpha$. The resulting logits are passed through a softmax function to produce a probability distribution over the next possible tokens. If $z$ does not belong to $\mathcal{V}_{\text{head}}(x_{<i})$, the probability of generating a token is set to zero. This contrastive decoding mechanism enhances the model's ability to reduce hallucinations by favoring contextually relevant token predictions over potentially misleading ones.

## 4 Experimentation Design

The experiment with the Delta method is conducted on a range of question-answering datasets and common-sense answering evaluations to assess its ability to mitigate hallucinations. Furthermore, the study presents several empirical observations to explore the characteristics of the Delta method when applied to the model used in this work.

### 4.1 Evaluation Datasets

To evaluate our method comprehensively, we selected diverse datasets that target various aspects of language model performance. These datasets are categorized based on their inclusion of context, question types, and the challenges they pose, providing a robust foundation for assessing the effectiveness of our approach.

The Stanford Question Answering Dataset (SQuAD) Rajpurkar et al. (2016) is widely used for training and evaluating machine reading comprehension models. SQuAD v1.1 contains over 100,000 question-answer pairs, with answers found directly in the text, while SQuAD v2 introduces over 50,000 unanswerable questions. This additional challenge makes SQuAD v2 particularly valuable for hallucination testing. Unanswerable questions allow us to assess if a model can correctly avoid generating answers when no relevant information is present.

TriviaQA is a large-scale question-answering dataset comprising over 650,000 question-answer pairs from trivia quizzes Joshi et al. (2017). It is more challenging than SQuAD because the answers can be spread across long and complex documents, including Wikipedia articles and web documents. The dataset evaluates models' ability to retrieve answers from longer, less structured texts. Google's Natural Questions (NQ) dataset Kwiatkowski et al. (2019) contains questions that users naturally ask in Google search queries. The answers are retrieved from long Wikipedia documents, and only a tiny portion of the document is directly relevant to the answer. This dataset tests models on open-domain question answering, requiring retrieval and comprehension of long papers with minimal direct context.

The four datasets mentioned above all include contextual information, and our method is expected to demonstrate significant improvements in them. In contrast, we also prepared two standard

question-answering datasets without context, where we anticipate limited performance gains from our method.

CommonsenseQA Talmor et al. (2019) is a dataset that evaluates a model's ability to answer questions requiring commonsense answering. It consists of multiple-choice questions, each testing the model's understanding of basic commonsense knowledge that cannot be easily derived from the text alone, requiring inference beyond surface-level information. MMLU (Massive Multitask Language Understanding) Hendrycks et al. (2021) is a comprehensive benchmark covering 57 subjects across various domains, such as humanities, STEM, social sciences, and more. MMLU is designed to assess a language model's knowledge across multiple disciplines, making it a robust test of general and subject-specific understanding of language models.

## 4.2 EXPERIMENTATION SET-UP

We utilize the Llama 3.1 8B Instruct model with 4-bit quantization as the baseline configuration Dettmers et al. (2023). The same model setup is applied for the Delta method, with parameters fixed at $r_{mask} = 0.7$, $\alpha = 0.3$, and $\beta = 0.1$ for all experiments. All experiments utilize the end-of-sequence (eos) token as the MASK token.

The experiments are divided into two categories: with sampling and without sampling. For the sampling experiments, the temperature is set to 1 to observe the impact of sampling on the Delta method's performance compared to non-sampling configurations.

| Dataset | Sample | Name | Exact Match | F1 | HasAns_EM | NoAns_EM |
|---|---|---|---|---|---|---|
| **SQuAD v1.1** | w/o | Baseline | 58.81741 | 72.37654 | - | - |
| | w/o | **Delta** | **61.81646** | **73.37708** | - | - |
| | w/ | Baseline | 57.51183 | 71.74116 | - | - |
| | w/ | **Delta** | **61.94891** | **73.39019** | - | - |
| **SQuAD v2** | w/o | Baseline | 41.32907 | 47.55297 | 59.07557 | 23.63331 |
| | w/o | **Delta** | **47.80595** | **52.94927** | 57.47301 | **38.16653** |
| | w/ | Baseline | 40.09096 | 46.47858 | 58.21525 | 22.01850 |
| | w/ | **Delta** | **46.20568** | **51.54408** | **58.62011** | **33.82675** |
| **TriviaQA** | w/o | Baseline | 48.27240 | - | - | - |
| | w/o | **Delta** | 48.12751 | - | - | - |
| | w/ | Baseline | 35.38787 | - | - | - |
| | w/ | **Delta** | **43.22893** | - | - | - |
| **Natural Question** | w/o | Baseline | 14.87535 | - | - | - |
| | w/o | **Delta** | 14.57064 | - | - | - |
| | w/ | Baseline | 9.25208 | - | - | - |
| | w/ | **Delta** | **11.80155** | - | - | - |

Table 1: Performance comparison between Baseline and Delta across various datasets.

## 5 RESULTS

To evaluate the effectiveness of the Delta method, we conducted comprehensive experiments on diverse QA datasets, including SQuAD versions 1.1 and 2, TriviaQA, and Natural Questions. The results, summarized in Table 1, highlight the performance improvements achieved by Delta across these benchmarks.

## 5.1 SQUAD V1.1 AND SQUAD V2

In SQuAD v1.1, the Delta method demonstrated its ability to enhance performance significantly, achieving exact match scores of 61.95 and 61.82 in experiments with and without sampling, respectively. These scores represent improvements of 4.44 and 3 percentage points over the baseline, underscoring the method's potential for refining model accuracy in extracting precise answers from

contextual data. In addition to exact match scores, F1 scores showed noticeable enhancements, further emphasizing the Delta method's robustness in handling contextual environments and reducing hallucinations.

Similarly, in SQuAD v2, which introduces a more challenging setting with unanswerable questions, the Delta method exhibited superior performance. The exact match scores surpassed the baseline by approximately six percentage points in both sampling and non-sampling scenarios, demonstrating the method's adaptability to different configurations. A particularly noteworthy result was observed in the "no answer" category, where the Delta method achieved remarkable improvements. For the exact match score of "no answer," the technique recorded increases of 14.53 and 11.81 percentage points for sampling and non-sampling setups, respectively. This indicates that the Delta method is especially effective when the context does not support a valid answer, highlighting its ability to mitigate hallucinations and prevent the generation of misleading information.

## 5.2 TRIVIAQA AND NATURAL QUESTIONS

The Delta method was evaluated on the TriviaQA and Natural Questions benchmarks to assess its effectiveness in context-aware question answering. The results demonstrated that increasing the sampling temperature significantly enhanced both baseline and Delta's performance, with improvements of 7.84 percent on TriviaQA and 2.55 percent on Natural Questions. However, Delta's progress was marginal without sampling, reflecting the datasets' complexity, such as multi-paragraph answer extraction in TriviaQA. The study suggests these results occur because sampling, by nature, is more prone to generating hallucinations due to the higher likelihood of sampling lower logit tokens. Since Delta reduces the logits of hallucinated tokens, it helps prevent them from being sampled, leading to better performance in tasks requiring more context-aware reasoning. This is why Delta shows more significant improvements in generation with sampling decoding.

| Name | CommonsenseQA (Acc) | MMLU (Acc) |
|---|---|---|
| Baseline | 75.51188 | 65.93078 |
| Delta | 75.26618 | 65.63880 |

Table 2: Accuracy comparison between Baseline and Delta models on CommonsenseQA and MMLU datasets.

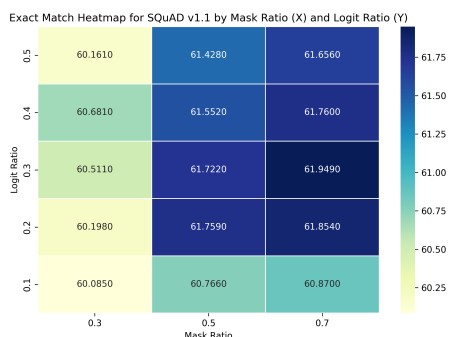

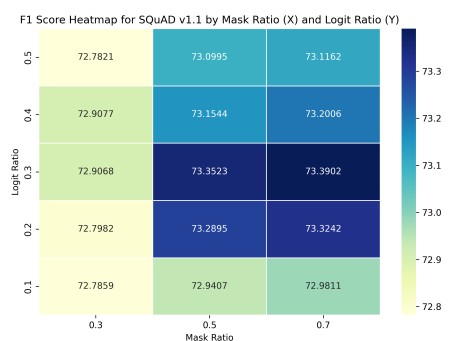

(a) Exact Match Heatmap for SQuAD v1.1        (b) F1 Score Heatmap for SQuAD v1.1

Figure 2: Comparison of Exact Match and F1 Score Heatmaps for Delta Model on SQuAD v1.1

## 5.3 COMMONSENSE QUESTION-ANSWERING AND MMLU

CommonsenseQA and MMLU are two question-answering benchmarks that differ from context-rich datasets in that they do not provide additional supporting context for the questions. Instead, models must rely entirely on the knowledge trained during pre-training to generate answers. This difference

limits the applicability of Delta's random masking approach, which focuses on contrasting masked contextual information to mitigate hallucinations.

As presented in Table 2, the evaluation results on these datasets show that Delta had marginal performance declines of 0.25 percentage points on CommonsenseQA and 0.29 percentage points on MMLU compared to the baseline. These small decreases indicate that Delta's masking mechanism, designed to work on context-dependent tasks, does not enhance performance when no external context is provided.

The result underscores Delta's vital limitation. While effective at reducing hallucinations and improving accuracy in tasks where context is explicitly available, its impact is minimal in context-free scenarios. This suggests that Delta is best suited for applications where contextual information is critical in guiding the model's predictions rather than tasks requiring innate knowledge or reasoning purely from pre-trained parameters.

## 6 ABLATION STUDY

In the ablation study, we investigated the effects of varying masking ratios and logit ratios ($\alpha$) on the overall performance of our method. These experiments were conducted on the SQuAD v1.1 dataset with sampling, using a temperature of 1 and $\beta = 0.1$ as the experimental setup. Masking ratios were set at 0.3, 0.5, and 0.7, while logit ratios ranged from 0.1 to 0.5. The results are summarized in heatmaps to provide a clear visualization of performance trends.

Figure 2 illustrates the heatmaps for exact match and F1 scores. The findings reveal minimal variation across different parameter settings, with standard deviations of 0.66 for exact match and 0.21 for F1 score. Notably, all parameter configurations achieved results that exceeded the baseline scores of 57.51 for exact match and 71.74 for F1. This highlights the robustness of the Delta method, demonstrating its ability to consistently perform well without requiring extensive hyperparameter tuning. These results emphasize the method's adaptability and reliability across various parameter values.

## 7 SUMMARY AND FUTURE WORKS

This study introduces Delta, an inference-time method designed to mitigate hallucinations in large language models without requiring additional fine-tuning. Delta operates by randomly masking input tokens to identify hallucination-prone logits and then subtracting these from the original logits, effectively reducing the influence of hallucinations. Experimental results demonstrate Delta's effectiveness in context-rich question-answering tasks, achieving significant performance improvements across datasets such as SQuAD, TriviaQA, and Natural Questions. However, its impact is limited in context-free tasks like CommonsenseQA and MMLU, where the model relies solely on pre-trained knowledge instead of external context. These findings position Delta as a powerful solution tailored for context-dependent tasks, offering valuable insights for reducing hallucinations in real-world applications.

Delta employs a straightforward random masking method, which has proven effective but leaves room for improvement. Future research will focus on developing advanced and adaptive masking strategies. One promising direction is targeted masking, prioritizing critical tokens such as proper nouns and key terms rather than applying masking uniformly. Additionally, leveraging techniques like part-of-speech tagging to prioritize tokens of higher informational value, such as nouns and verbs, could refine the method further. These approaches could enhance Delta's adaptability, making it more robust across diverse QA scenarios.

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
