# OpenReview forum: "Delta - Contrastive Decoding Mitigates Text Hallucinations in Large Language Models"
_ICLR.cc/2025/Conference — Submitted to ICLR 2025_

### Official Review · Reviewer_R3me · 2024-10-27

**Soundness:** 3
**Presentation:** 3
**Contribution:** 2
**Rating:** 5
**Confidence:** 4

**Summary:**

This paper introduces Delta, an inference-time approach designed to reduce hallucinations in LLMs without requiring model retraining or additional data. Delta utilizes contrastive decoding by masking parts of the input prompt, and then comparing the output distributions generated from masked and unmasked inputs. Through this mechanism, Delta aims to detect and mitigate hallucinations effectively. The method was evaluated on multiple benchmark datasets, including SQuAD v1.1 and v2, and showed performance improvements of up to 6% in hallucination-prone scenarios.

**Strengths:**

1. The paper is easy to follow.
2. The method is computationally efficient, making it suitable for integration into existing systems where minimal adjustments are preferable.
3. The experimental setup on SQuAD and other datasets is well-executed.

**Weaknesses:**

1. In this method, the tokens to be masked are selected randomly, which may lead to inconsistent performance.
2. Delta’s performance declines in tasks that rely more on pre-trained knowledge than contextual cues, such as CommonsenseQA and MMLU. This limits its applicability in domains that require factual correctness without strong context-based cues.
3. The method's effectiveness appears sensitive to hyperparameter settings, including mask and logit ratios. This limits the method's practicality because it requires setting different hyperparameters for different tasks.

**Questions:**

1. Have you evaluated the performance of this method with different hyperparameter combinations on lower-performing benchmarks, such as CommonsenseQA and MMLU?

---

> ### Author Response · Authors · 2024-11-20
> **Reply to the comments from the Reviewer R3me**
>
> First, the research team appreciates your time and comments on our work. For the first concern, the masking tokens are selected randomly to maximize generalizability, ensuring applicability across diverse contexts and prompts without being tailored to specific cases. Based on the ablation study results shown in Figure 2, our method exhibits low sensitivity to hyperparameter variations, with standard deviations of less than 0.7 for Exact Match and 0.3 for F1 scores. This addresses your concern regarding sensitivity to different settings. Additionally, the results demonstrate that all tested configurations outperform the baseline model without the Delta method.
>
> Regarding the concerns about the CommonsenseQA and MMLU datasets, the primary objective of our method is to showcase its effectiveness in context-driven tasks. Evaluations that rely heavily on pre-trained knowledge, such as those in CommonsenseQA and MMLU, are not the core focus of this approach.

---

### Official Review · Reviewer_EqRA · 2024-11-01

**Soundness:** 2
**Presentation:** 2
**Contribution:** 2
**Rating:** 3
**Confidence:** 4

**Summary:**

This paper introduces Delta, an inference-time approach to mitigate hallucinations in large language models (LLMs) without requiring model retraining. The method leverages contrastive decoding by randomly masking portions of input prompts and comparing the output distributions of masked and unmasked inputs. By dynamically adjusting logits during token generation, Delta aims to reduce hallucinated content, particularly in context-driven question-answering tasks. Experimental results on datasets like SQuAD demonstrate significant improvements, with the approach showing particular strength in handling ambiguous contexts and unanswerable questions, achieving up to 14.56% improvement in exact match accuracy for scenarios with no definitive answers.

**Strengths:**

- Delta provides an efficient, inference-only solution to reduce hallucinations, avoiding retraining and keeping implementation straightforward.

- The method shows some promising result in improvement question-answering tasks.

**Weaknesses:**

- The text sequence masking is currently using random masking. This is a very basic approach. A better approach would be masking important tokens (e.g. entities).
- Although Delta performs well in ambiguous question-answering datasets like SQuAD v1.1 and v2, its effectiveness is less pronounced in tasks like TriviaQA and Natural Questions, where factual recall or logical inference is needed.
- The exact match and F1 score improvements in CommonsenseQA and MMLU are marginal, which may limit Delta's broader application.
- Delta is proposed to mitigate hallucination, however, it is evaluated on QA datasets. The method should be extended to other tasks for example, summarization, where hallucination is known to be a problem.
- The baseline in the paper is just the vanilla Llama 3 model. Delta should be compared with other baselines with similar approach which is missing in the current work:
  1. Contrastive decoding using multiple input contexts (including relevant and irrelevant) [1];
  2. Context aware decoding (CAD) [2]



**Reference**

[1] Zhao et al. Enhancing Contextual Understanding in Large Language Models through Contrastive Decoding

[2] Shi et al. Trusting Your Evidence: Hallucinate Less with Context-aware Decoding

**Questions:**

1. Have the authors considered an adaptive masking mechanism where the degree of masking adjusts based on the input’s complexity or ambiguity level? In addition, masking important tokens (e.g. entity) would be a better alternative than random masking. There is a concurrent work with very similar idea [1].
2. Table 1 and Figure 2 are quite duplicated. Why is Figure 2 needed?
3. The study uses the Llama 3.1 8B model with 4-bit quantization, which may not be the most competitive model. Had the authors considered conducting further experiments using larger models?



**Reference**

[1] Gema et al. DECORE: DECODING BY CONTRASTING RETRIEVAL HEADS TO MITIGATE HALLUCINATIONS

---

> ### Author Response · Authors · 2024-11-26
>
> We appreciate your time and comments on our work.
>
> - Regarding weakness 1: While random masking is indeed an intuitive approach, our work aims to demonstrate its effectiveness. The CD method could mitigate hallucination in LLMs. Using important token masking would be a promising direction for future work.
> - Regarding weakness 2: We have conducted experiments for Delta with decoding sampling (temperature set to 1) and found improved performance across all four datasets with decoding sampling. The updated table is shown below and will be incorporated into the paper.
> - Regarding weakness 3: You are correct that the Delta method shows limited improvements in tasks without context. This is because the technique is designed to enhance task performance with contextual information, effectively reducing hallucinations and improving reliability. For tasks without context, the benefits of Delta are naturally less pronounced.
> - Regarding weakness 4. This work aims to focus on context-driven question answering to investigate the impact of hallucination on LLM and the degree of mitigation using our Delta method.
> - Regarding weakness 5: As for your last concern, the study has updated the related work section to clarify the difference between the Delta methods for hallucination mitigation and the CAD method for the relevant datasets.
>
> ### **In response to your questions:**
>
> - Question 1: Yes, we considered masking entity tokens. However, this approach would require an NLP model like BERT to identify entities in text, increasing computational resources during inference. Alternative methods like regular expressions would limit Delta's applications (making it task-specific). We, therefore, chose the basic method for our experiments.
> - Question 2: We have removed the duplicated figure 2 in the updated paper.
> - Question 3: Yes, we plan to conduct further experiments with larger models in future work to evaluate the scalability and performance of our method on more competitive architectures.
>
> Thank you for the suggestion. We agree that comparing our method with other methods would provide valuable insights. Due to time constraints, we could not conduct additional experiments for this submission. However, we plan to include such comparisons in future work to validate our method further and position it within the broader landscape of existing approaches.
>
> | **Dataset**         | **Sample** | **Name**       | **Exact Match** | **F1**       | **HasAns_EM** | **NoAns_EM** |
> |----------------------|------------|----------------|------------------|--------------|---------------|--------------|
> | **SQuAD v1.1**       | w/o        | Baseline       | 58.81741         | 72.37654     | -             | -            |
> |                      | w/o        | **Delta**      | **61.81646**     | **73.37708** | -             | -            |
> |                      | w/         | Baseline       | 57.51183         | 71.74116     | -             | -            |
> |                      | w/         | **Delta**      | **61.94891**     | **73.39019** | -             | -            |
> | **SQuAD v2**         | w/o        | Baseline       | 41.32907         | 47.55297     | 59.07557      | 23.63331     |
> |                      | w/o        | **Delta**      | **47.80595**     | **52.94927** | 57.47301      | **38.16653** |
> |                      | w/         | Baseline       | 40.09096         | 46.47858     | 58.21525      | 22.01850     |
> |                      | w/         | **Delta**      | **46.20568**     | **51.54408** | **58.62011**  | **33.82675** |
> | **TriviaQA**         | w/o        | Baseline       | 48.27240         | -            | -             | -            |
> |                      | w/o        | **Delta**      | 48.12751         | -            | -             | -            |
> |                      | w/         | Baseline       | 35.38787         | -            | -             | -            |
> |                      | w/         | **Delta**      | **43.22893**     | -            | -             | -            |
> | **Natural Question** | w/o        | Baseline       | 14.87535         | -            | -             | -            |
> |                      | w/o        | **Delta**      | 14.57064         | -            | -             | -            |
> |                      | w/         | Baseline       | 9.25208          | -            | -             | -            |
> |                      | w/         | **Delta**      | **11.80155**     | -            | -             | -            |

---

> > ### Comment · Reviewer_EqRA · 2024-11-27
> >
> > Thank you authors for the reply. After reading the response and the updated paper, I decide to maintain my score.

---

### Official Review · Reviewer_9X2z · 2024-11-04

**Soundness:** 2
**Presentation:** 3
**Contribution:** 2
**Rating:** 3
**Confidence:** 4

**Summary:**

This paper introduces using constrastive decoding to mitigate hallucinations in LLM. The so-called Delta method works by randomly masking the input text and adjusting predictions using the masked output distribution. Experiments are conducted on several well-known benchmarks, such as SQuAD, TriviaQA, NQ, CommonsenseQA, MMLU etc. The method brings +4~6 EM scores on SQuAD and no improvements on other benchmarks. The paper is generally well-written and easy to follow.

**Strengths:**

Reducing LLM hallucinations is an import research topic and challenging. This paper tries to introduce constrastive decoding (CD) to tackle this problem. The motivation is intuitive and reasonable as CD has been verifyed in other tasks. Different from CD based on Gaussian noise etc. in visual tasks, this paper adopts token-level masking in text LLM. Experiments on SQuAD show the proposed CD method significantly improves the accuracy.

**Weaknesses:**

Experiments are conducted on several benchmarks. However, the proposed method only works on SQuAD and shows no improvements on otherss. Therefore, the generalization capability of the proposed method is limited. Why the proposed method does not work well on others tasks is unknown as well.

In addition to visual tasks, CD is also used in other NLP tasks, such as  "Contrastive Decoding Reduces Hallucinations in Large Multilingual Machine Translation Models", "Mitigating Hallucinations and Off-target Machine Translation with Source-Contrastive and Language-Contrastive Decoding" etc..   Whether these existing methods is effective in LLM is unknown. It's better to compare these methods  (or some) with the proposed one to verfiy its effectiveness.

Ablation studies on technique choices are missing. For example, why chosing masking tokens instead of other variations? what's the best masking strategy?  how the multiply factors in  eq(3) are determined? Why the APC is necessary and how to choose the beta value? etc.

**Questions:**

see weaknesses.

---

> ### Author Response · Authors · 2024-11-20
> **Reply to the comment from the Reviewer 9X2z**
>
> First, the research team appreciates your time and comments on our work. For the first question, we have updated experiments by facilitating sampling using higher temperatures; the results have shown improvement from natural questions to TriviaQA. The detailed results are available in the table below. As for the ablation study concerns, we conducted a complete experiment on various combinations of hyperparameters and their effect on experiment results in the SQuAD dataset, which is available in Figure 3. In addition, we will have a more detailed discussion in the new section of this paper to understand this method in more detail.
>
> | **Dataset**         | **Sample** | **Name**       | **Exact Match** | **F1**       | **HasAns_EM** | **NoAns_EM** |
> |----------------------|------------|----------------|------------------|--------------|---------------|--------------|
> | **SQuAD v1.1**       | w/o        | Baseline       | 58.81741         | 72.37654     | -             | -            |
> |                      | w/o        | **Delta**      | **61.81646**     | **73.37708** | -             | -            |
> |                      | w/         | Baseline       | 57.51183         | 71.74116     | -             | -            |
> |                      | w/         | **Delta**      | **61.94891**     | **73.39019** | -             | -            |
> | **SQuAD v2**         | w/o        | Baseline       | 41.32907         | 47.55297     | 59.07557      | 23.63331     |
> |                      | w/o        | **Delta**      | **47.80595**     | **52.94927** | 57.47301      | **38.16653** |
> |                      | w/         | Baseline       | 40.09096         | 46.47858     | 58.21525      | 22.01850     |
> |                      | w/         | **Delta**      | **46.20568**     | **51.54408** | **58.62011**  | **33.82675** |
> | **TriviaQA**         | w/o        | Baseline       | 48.27240         | -            | -             | -            |
> |                      | w/o        | **Delta**      | 48.12751         | -            | -             | -            |
> |                      | w/         | Baseline       | 35.38787         | -            | -             | -            |
> |                      | w/         | **Delta**      | **43.22893**     | -            | -             | -            |
> | **Natural Question** | w/o        | Baseline       | 14.87535         | -            | -             | -            |
> |                      | w/o        | **Delta**      | 14.57064         | -            | -             | -            |
> |                      | w/         | Baseline       | 9.25208          | -            | -             | -            |
> |                      | w/         | **Delta**      | **11.80155**     | -            | -             | -            |

---

### Official Review · Reviewer_JQk7 · 2024-11-04

**Soundness:** 1
**Presentation:** 1
**Contribution:** 1
**Rating:** 1
**Confidence:** 5

**Summary:**

The paper proposes a constructive decoding method to reduce hallucinations, which compares the original logits with the logits computed from corrupt inputs.

**Strengths:**

The method is intuitive and easy to implement.

**Weaknesses:**

1. The paper lacks plenty of related works and baselines. I suggest the author at least compare the following de-hallucination baselines:
    + CAD [1]
    + ITI [2]
    + DoLa [3]
    + AD [4]
    + SEA [5]
2. The method and equation are similar to CAD [1], where the masking part is the corresponding context for ODQA tasks. However, the paper does not mention and compare this method in SQuAD v1.1/v2, TriviaQA, and Natural Question.
3. The random masking leads to incompatible with FlashAttention, preventing its practical applications.
4. The performance improvement of the method is not significant, e.g., a negative improvement on TriviaQA and Natural Question, 0.25 and 0.3 improvements on CommensenseQA and MMLU
5. The paper is not well written.

[1] Trusting your evidence: Hallucinate less with context-aware decoding

[2] Inference-time intervention: Eliciting truthful answers from a language model

[3] Dola: Decoding by contrasting layers improves factuality in large language models

[4] In-context sharpness as alerts: An inner representation perspective for hallucination mitigation

[5] Spectral Editing of Activations for Large Language Model Alignment

**Questions:**

No further questions.

---

> ### Author Response · Authors · 2024-11-26
>
> We appreciate your time and comments on our work.
>
> - Regarding weakness 1: We have studied methods like CAD to investigate the differences among these works. The detailed results are available in the related works and the result sections.
> - Regarding weakness 2: Regarding your concerns about weakness 2, we compared the CAD and Delta methods in the related work section to discuss their capability for hallucination mitigation in context-driven tasks.
> - Regarding weakness 3, our method only masks input tokens and operates on the overall output logits during inference without interfering with internal attention mechanisms. Therefore, it remains fully compatible with FlashAttention and does not impact its functionality. We have conducted an experiment with FlashAttention 2 enabled, which worked correctly.
> - Regarding weakness 4: We have conducted experiments for Delta with decoding sampling (temperature set to 1) and found improved performance across all four datasets with decoding sampling. The updated table is shown below and will be incorporated into the paper.
>
> | **Dataset**         | **Sample** | **Name**       | **Exact Match** | **F1**       | **HasAns_EM** | **NoAns_EM** |
> |----------------------|------------|----------------|------------------|--------------|---------------|--------------|
> | **SQuAD v1.1**       | w/o        | Baseline       | 58.81741         | 72.37654     | -             | -            |
> |                      | w/o        | **Delta**      | **61.81646**     | **73.37708** | -             | -            |
> |                      | w/         | Baseline       | 57.51183         | 71.74116     | -             | -            |
> |                      | w/         | **Delta**      | **61.94891**     | **73.39019** | -             | -            |
> | **SQuAD v2**         | w/o        | Baseline       | 41.32907         | 47.55297     | 59.07557      | 23.63331     |
> |                      | w/o        | **Delta**      | **47.80595**     | **52.94927** | 57.47301      | **38.16653** |
> |                      | w/         | Baseline       | 40.09096         | 46.47858     | 58.21525      | 22.01850     |
> |                      | w/         | **Delta**      | **46.20568**     | **51.54408** | **58.62011**  | **33.82675** |
> | **TriviaQA**         | w/o        | Baseline       | 48.27240         | -            | -             | -            |
> |                      | w/o        | **Delta**      | 48.12751         | -            | -             | -            |
> |                      | w/         | Baseline       | 35.38787         | -            | -             | -            |
> |                      | w/         | **Delta**      | **43.22893**     | -            | -             | -            |
> | **Natural Question** | w/o        | Baseline       | 14.87535         | -            | -             | -            |
> |                      | w/o        | **Delta**      | 14.57064         | -            | -             | -            |
> |                      | w/         | Baseline       | 9.25208          | -            | -             | -            |
> |                      | w/         | **Delta**      | **11.80155**     | -            | -             | -            |

---

### Official Review · Reviewer_EmNp · 2024-11-08

**Soundness:** 2
**Presentation:** 2
**Contribution:** 1
**Rating:** 3
**Confidence:** 5

**Summary:**

This work aims at proposing an inference-time improvement to the regular decoding algorithms by leveraging contrastive-based decoding. Specifically, the authors propose Delta that contrastively decodes from two input contexts, an original context and a masked context with part of the input removed. Experiments in question answering show that the method can help mitigate hallucinations.

**Strengths:**

This work tries to tackle an important and interesting problem of reducing regular LLMs' hallucination by modifying the decoding algorithm at inference time. The experiments show good results on SQuAD. The writing is overall clear and structured.

**Weaknesses:**

(1) This work misses a reference and comparison to context-aware decoding (Shi et al., 2023). The two methods seem very similar (e.g., Equation (5) in this work is similar to Section 2.2 in Shi et al., where a partial context is removed contrastively). The authors should consider adding a comparison of the two methods' performance and discuss any potential differences between the two.

(2) This work seems to be experimenting with only one LLM, specifically LLama 3.1 8B. Different model families and different model sizes should be reported. Also, the used Llama 3.1 8B is a non-instructed version. It would be interesting to see experiments with e.g., Llama-3.1-8B-Instruct.

(3) This work only shows performance improvements in SQuAD. In TriviaQA, Natural Question, CommonsenseQA, and MMLU, the proposed Delta method is worse than the baseline decoding method.

(4) The mask(z) operation could benefit from exploring more design choices. For example, how would using tokens other than EOS work as the mask token? How would transformations other than masking work (e.g., reordering, summarization/back-translation, randomly copying in irrelevant contexts, etc.)?

References
Shi et al., 2023. Trusting Your Evidence: Hallucinate Less with Context-aware Decoding. https://arxiv.org/abs/2305.14739

**Questions:**

N/A

---

> ### Author Response · Authors · 2024-11-20
> **Reply to the comment from Reviewer EmNp**
>
> First, we are grateful for your insightful comments and suggestions. They have been instrumental in improving our work. We have updated our paper to include new empirical evidence demonstrating notable improvements across various datasets, including natural questions and triviaQA. Additionally, we clarified the model specification, explicitly stating that our model is LLaMA 3.1-8B-Instruct rather than the original base model. This update addresses an oversight caused by insufficient contextual details in our experimental descriptions.
>
> Regarding Question 1, we are revising the paper to include the latest theoretical comparisons and a more detailed discussion of the key differences in our methodology. Per your question 3, we have conducted a new experiment that facilitates sampling with our method, which has consistently yielded better results from TriviaQA to Natural Questing datasets. For question 4, the central hypothesis of our work emphasizes achieving high generalization. This allows us to utilize a wide range of “meaningless” tokens, such as EOS or empty string tokens, for masking without distorting or reorganizing the content's structure.
>
> The updated results are available in the table below.
>
>
> | **Dataset**         | **Sample** | **Name**       | **Exact Match** | **F1**       | **HasAns_EM** | **NoAns_EM** |
> |----------------------|------------|----------------|------------------|--------------|---------------|--------------|
> | **SQuAD v1.1**       | w/o        | Baseline       | 58.81741         | 72.37654     | -             | -            |
> |                      | w/o        | **Delta**      | **61.81646**     | **73.37708** | -             | -            |
> |                      | w/         | Baseline       | 57.51183         | 71.74116     | -             | -            |
> |                      | w/         | **Delta**      | **61.94891**     | **73.39019** | -             | -            |
> | **SQuAD v2**         | w/o        | Baseline       | 41.32907         | 47.55297     | 59.07557      | 23.63331     |
> |                      | w/o        | **Delta**      | **47.80595**     | **52.94927** | 57.47301      | **38.16653** |
> |                      | w/         | Baseline       | 40.09096         | 46.47858     | 58.21525      | 22.01850     |
> |                      | w/         | **Delta**      | **46.20568**     | **51.54408** | **58.62011**  | **33.82675** |
> | **TriviaQA**         | w/o        | Baseline       | 48.27240         | -            | -             | -            |
> |                      | w/o        | **Delta**      | 48.12751         | -            | -             | -            |
> |                      | w/         | Baseline       | 35.38787         | -            | -             | -            |
> |                      | w/         | **Delta**      | **43.22893**     | -            | -             | -            |
> | **Natural Question** | w/o        | Baseline       | 14.87535         | -            | -             | -            |
> |                      | w/o        | **Delta**      | 14.57064         | -            | -             | -            |
> |                      | w/         | Baseline       | 9.25208          | -            | -             | -            |
> |                      | w/         | **Delta**      | **11.80155**     | -            | -             | -            |

---

### Meta-Review · Area_Chair_xWjp · 2024-12-20

**Metareview:**

The paper introduces "Delta", an inference-time method aimed at reducing hallucinations in large language models (LLMs) via contrastive decoding. The core idea is to decode from both original and masked contexts and contrast the results to mitigate hallucinations. The method is tested on multiple benchmarks like SQuAD, TriviaQA, and others.

The paper has several flaws:
1. A notable gap is the lack of comparison with directly similar methods, especially those using contrastive decoding strategies. The cited CAD method, along with others such as ITI and SEA, should be thoroughly compared to underscore Delta's unique contributions.
2. The paper's results on datasets like TriviaQA and MMLU are less promising. It is paramount to analyze why these other datasets do not see similar gains and whether the method inherently struggles with tasks requiring factual accuracy or logical inference.
3. While the authors claim low sensitivity, an explicit discussion and expanded ablation study providing insights into how different masking strategies or other hyper-parameters affect performance across a variety of tasks would be beneficial.
4. The manuscript can benefit from additional detail and clarity, particularly in the method description section.
5. As suggested by reviewers, experimenting with different model sizes, particularly on instructed models (e.g., LLaMA-3.1-8B-Instruct), and extending the study beyond QA datasets to tasks like summarization, can improve the comprehensiveness of the results.

**Additional Comments On Reviewer Discussion:**

The reviewer discussions mainly highlighted the need for broader comparisons with existing methods and raised concerns about generalization across datasets. The authors attempted to address these by running additional experiments and updating comparisons, though some reviewers felt the comparisons still lacked depth.

---

### Decision · Program_Chairs · 2025-01-22

Reject